# Further Insights on Predictors of Environmental Tobacco Smoke Exposure during the Pediatric Age

**DOI:** 10.3390/ijerph16214062

**Published:** 2019-10-23

**Authors:** Carmela Protano, Vittoria Cammalleri, Arianna Antonucci, Alexandra Sabina Ungureanu, Francesa Santilli, Stefano Martellucci, Vincenzo Mattei, Matteo Vitali

**Affiliations:** 1Department of Public Health and Infectious Diseases, Sapienza University of Rome, 00185 Rome, Italy; carmela.protano@uniroma1.it (C.P.); vittoria.cammalleri@uniroma1.it (V.C.); arianna.antonucci@uniroma1.it (A.A.); Alexandra.u@libero.it (A.S.U.); 2Biomedicine and Advanced Technologies Rieti Center, “Sabina Universitas”, 02100 Rieti, Italy; francesca.santilli@uniroma1.it (F.S.); stefano.martellucci@uniroma1.it (S.M.); vincenzo.mattei@uniroma1.it (V.M.); 3Department of Experimental Medicine, Sapienza University of Rome, 00185 Rome, Italy

**Keywords:** Environmental Tobacco Smoke, home smoking policies, smoking ban, ethnicity, educational level, children

## Abstract

*Background:* The smoking ban in public places has reduced Environmental Tobacco Smoke (ETS) exposure for non-smokers, but despite this, domestic environments still remain places at high risk of exposure, and, today, about 40% of children worldwide are exposed to ETS at home. The aims of the study are to investigate the contribution of several factors on ETS exposure among a group of Italian children and to evaluate the changes in smoking precautions adopted at home when the smoker is the mother, the father, or both parents, respectively. *Methods:* A cross-sectional study was performed on a sample of 519 Italian schoolchildren. Information was collected via a questionnaire. *Results:* 41.4% of the participants lived with at least one smoker. Almost half of the children exposed to ETS lived with one or more smokers who do not observe any home smoking ban. Lower maternal or paternal educational levels significantly increase the risk of ETS exposure at home and the “worst case” is represented by both parents who smoke. *Conclusions:* More effective preventive interventions are needed to protect children from ETS exposure. Some interventions should be specifically dedicated to smokers with a low educational level and to mothers that smoke.

## 1. Introduction

Exposure to passive smoking, also called Environmental Tobacco Smoke (ETS) exposure, has been associated with a great number of adverse health effects on non-smokers, including cardiovascular, respiratory, and neoplastic diseases [1,2,3]. Furthermore, ETS exposure of infants and children, who are known to be particularly vulnerable, has been linked with several negative outcomes that can occur during childhood and/or later in life during adulthood. Indeed, recent research demonstrated the association between the ETS exposure of children and respiratory tract problems, such as asthma, persistent wheezing, changes in lung growth and development, and respiratory tract infections [4,5]. Further scientific evidence highlighted an increased risk of childhood leukemia related to ETS exposure during pregnancy, alone, or combined with postnatal exposure of newborns [6,7]. Finally, ETS exposure during childhood can contribute to the onset of adult diseases, especially pulmonary and cardiovascular ones [8].

Given all of this evidence, non-smokers should be fully protected from ETS exposure, especially the most susceptible populations, such as children. Over the last decades, it has been demonstrated that ETS results from the combination of two phenomena: the secondhand smoke (SHS) and the thirdhand smoke (THS) [9]. Exposure to SHS is the involuntary inhalation of smoke air pollutants that occurs when a subject is close to an active smoker during smoking [10,11,12], while exposure to THS occurs through the inhalation of indoor smoke air pollutants long after the cigarette has been extinguished [13]. Thus, indoor environments represent high-risk places for ETS exposure, including both environments where one or more individuals are smoking and environments in which one or more individuals have previously smoked [14]. ETS exposure risk has been partially managed in many countries, including Italy, by the introduction of restrictions or complete bans on smoking in public areas. Nevertheless, this kind of ban does not protect the health of non-smokers in private places, such as domestic environments, if they live with one or more smokers. This threat is of particular concern for children’s health, due to the longer time they spend at home with respect to the adults and the remarkable vulnerability typical of the pediatric age, and needs to be targeted for future studies [15]. In our previous research, performed on pediatric populations, we demonstrated that urinary (u-) cotinine levels (a well-known biological marker of ETS exposure) significantly increase in children with one or more cohabitant smokers [16,17]. In addition, we found that the u-cotinine levels of Italian children were strictly related to the smoking habits of cohabitant smoker(s), together with the smoking precautions adopted at home. In particular, u-cotinine levels significantly increased in a linear way as domestic ETS exposure increased. Briefly, we found the lowest levels among children living with non-smoking cohabitants, higher concentrations for children living with smokers that never smoke at home, levels a little higher for children living with smokers that smoke at home only when children are out, and the highest levels among children living with cohabitants that smoke at home also when children are present [16]. These results are consistent with the occurrence of exposure to SHS alone or in combination with THS. The ETS exposure profile of children in domestic environments changes also in relationship with some characteristics of the children themselves and their parents, such as ponderal status according to body mass index of children, paternal educational level [17,18,19], family income [20], parental status (e.g., single mother), use of prenatal preventive care, and parenting satisfaction [21]. Furthermore, in recent years, several studies focalized the attention on the contribution of smoking fathers or mothers to the exposure of cohabiting children to ETS [22,23].

The knowledge of the factors that contribute to the ETS exposure of children and to its “intensity” and the investigation of the habits of the smoking mothers or fathers are of particular importance to manage the related health risks, considering that domestic environments are the main source of exposure to ETS during pediatric age and that it is estimated that about 40% of children worldwide are exposed to ETS at home [24].

The aims of the present study were: (1) to investigate the association between several factors and ETS exposure among a group of healthy children living in Italy, and (2) to evaluate the changes in smoking precautions adopted at home when the smoker is the mother, the father or both parents, respectively.

## 2. Materials and Methods

### 2.1. Study Population and Design

A cross-sectional study was performed in the academic year 2017–2018 on a sample of 519 healthy schoolchildren (5–11 years old) attending three primary school districts located in Central Italy (Rieti Province, Latium Region). Details on the recruitment strategy and on the characteristics of the selected areas have been reported previously [25,26,27,28,29,30].

The research project was explained to all the students and their parents. Then, all the material for participating in the study (informed consent for parents and for children aged ≥9 years, the processing of personal data form and an ad hoc questionnaire) was delivered to the children. The families who decided to participate handed back the fulfilled documents to school on a date already agreed.

The protocol of the present study and all the documents necessary to participate were approved by the Ethics Committee of the teaching hospital Policlinico Umberto I of Rome, Italy (Protocol n. 2894/12.09.2013).

Information about children and their parents were collected by the use of the questionnaire, filled in by parents together with the informed consent and the processing of the personal data form. Questions on each participant explored gender, birth date, height, weight, nationality, and exposure to ETS in domestic environments (cohabitant smoker(s) and her/his/their habits at home). Questions on the parents referred to nationality and educational level.

### 2.2. Covariates Gathered by the Questionnaires and Statistical Analysis

All the information derived from the questionnaires and used for data analysis were coded and added in a database, specifically elaborated for statistical purposes. First of all, gender was coded as male = 0 and female = 1. The nationality of the child was coded as Italian = 0 and not Italian = 1.

ETS exposure status in the domestic environment was defined based on the presence of at least one cohabitant smoker; the answers were coded as follows: no = 0 and yes = 1. When the answer was “yes”, we considered the child exposed to ETS and, in this case, the parent who dealt with the questionnaire had to respond to a specific part concerning the smoke habits of any cohabitant smoker. Firstly, they were asked to indicate the number of cohabiting smokers; the response was coded as a dichotomous variable: only one smoker = 0 and >one smoker = 1. Then, we asked who the cohabitant smoker(s) is/are (mother, father, and/or others). For each cohabitant smoker, we required the number of cigarettes smoked in total, at home generally, and at home when the child is present, every day. In addition, we assessed if the cohabitant smoker(s) smoke at home and, in this case, if she/he/they smoke at home when the child is present. For each question, the possible answers were coded as: no = 0 and yes = 1. According to the responses, we generated a new variable, “Home smoking policy”, considering a total smoking ban when the cohabitant smoker(s) do not smoke at home (coded as 0), partial smoking ban when cohabitant smoker(s) smoke at home only when the child is out (coded as 1), and no smoking ban when cohabitant smoker(s) smoke at home also when the child is present (coded as 2).

The educational level of each parent was investigated through a closed-ended question. According to the Organization for Economic Co-operation and Development [31], the respective answers were codified as follows: 1, basic (≤9 years); 2, upper secondary; and 3, tertiary/higher.

For the statistical analyses, these groups have been further aggregated as follows: “upper secondary or lower” or “tertiary/higher”.

### 2.3. Statistical Analysis

Statistical data processing was performed using IBM SPSS Statistics 25 software (IBM Corp., Armonk, NY, USA). In descriptive statistics, categorical variables were reported as absolute and relative frequencies and continuous variables were reported as arithmetic mean ± Standard Deviation (SD). Differences in the prevalence of smoking habit depending on the nationality and the educational level of the father and of the mother were tested by the use of the chi-squared test. Then, considering only the sub-population of children exposed to ETS, the chi-squared test was used to evaluate possible differences in the prevalence of home smoking policies applied by cohabitant smoker(s) in domestic environments, also according to maternal or paternal nationality and educational levels. The same test was also used to assess possible differences in the application of a smoking ban when the cohabitant smoker was the mother, the father, or both parents. *p*-values were based on two-sided tests and significance was set at *p*-value < 0.05.

## 3. Results

In total, 379 children aged 5–11 years (51% female) participated in the research, resulting in a very high participation rate for this type of study, equal to 73%. Table 1 shows the characteristics of the study population and of the participants exposed to ETS.

Even in the sub-population of children exposed to ETS, the sample was similar in terms of gender (52.3% of females). Also, the percentages related to maternal and paternal nationality and educational level were similar to those found for the entire sample. Regarding the nationality of the parents, approximately 15% of the mothers and 13% of the fathers were not Italian, almost all from other European countries (Romania, Poland, and Albania). With respect to the parental education level, a great percentage of mothers and fathers had at least an upper secondary education, with a higher proportion for mothers (81.0% vs. 68.9%). Results concerning ETS exposure showed that 41.1% of the participants lived with at least one cohabitant smoker.

Approximately two-thirds of the participants included in this sub-population live with one smoker (in 44.1% of the cases it was the mother, in 51% of the cases it was the father, in 3.9% of the cases it was the grandmother, and in 1.9% of the cases it was the uncle); the remaining one-third live with two or more cohabitant smokers (always the mother and the father, in only one case the mother and the grandmother). A smaller percentage (3.9%) live with three cohabitant smokers (always the mother and the father and, in addition, the grandmother or the uncle). Finally, one of the participants lives with four cohabitant smokers (the mother, the father, the grandfather, and the grandmother).

Based on the home smoking policy adopted at home by cohabitant smokers, almost half of the children (38.7%) live with one or more smokers who do not observe a total smoking ban at home, and about 19% of the total participants live in domestic environments without any smoking ban. The mean of the cigarettes smoked in total, at home, and at home when the child was present were, respectively, about 16/day, 3/day, and 1/day.

Figure 1 summarizes the prevalence of smoking habits according to the nationality and the educational level of the mother or father.

As reported in Figure 1, the percentages of cohabitant smoker(s) were similar in the case of both Italian or not Italian mothers and fathers. In contrast, we found an inverse significant relationship between the degree of maternal and paternal educational level and the smoking habits (*p* = 0.022 and *p* < 0.001, respectively).

In Figure 2, the proportion of smoking bans applied at home of participants living with one or more cohabitant smokers according to the maternal and paternal nationality and educational level are reported: no statistically significant differences were found.

Focusing the attention on the ETS exposure profile and smoking habits of cohabitant smokers when the smoker was the participant’s mother and/or father, the results showed different situations of exposure and a different application of the smoking ban in domestic environments. In particular, we recognized three different ETS exposure profiles when the cohabitant smoker was only the mother (case 1), when the cohabitant smoker was only the father (case 2), and when mother and father were both smokers (case 3). Figure 3 summarizes the results related to ETS exposure profile and smoking habits-home smoke policies according to each of the three cases.

The results demonstrated that the worst case “no smoking ban” was found in 20% of cases in which the smoker was the mother, with respect to 7% of cases in which the cohabitant smoker was the father. Additionally, when the mother and the father were both smokers, the percentage of the cases in which no smoking ban was applied increased to 25.6%.

Table 2 shows the results of the logistic regression analysis performed to evaluate the contribution of parental nationality and educational level on ETS exposure among monitored schoolchildren.

The results of the multivariate analysis confirm the independent contribution of parental educational level on ETS exposure (maternal and paternal with a *p*-value equal to 0.046 and 0.001, respectively).

Table 3 shows the results of logistic regression analysis carried out on the ETS-exposed subgroup to assess the contribution of parental nationality and educational level on the application or not of an at-home smoking ban.

As reported in Table 3, no variables entered in the model, confirming also in this case the results of univariate analyses.

## 4. Discussion

ETS exposure of the general population remains one of the most important threats for human health. Indeed, despite the great number of negative outcomes associated with this exposure, especially when it occurs during pregnancy, early in life, or during childhood, the percentage of smokers is still high worldwide.

The first relevant finding of the present research is related to the prevalence of children exposed to ETS in domestic environments: we found that more than 40% of the participants live with at least one smoker. This is in line with the frequencies of exposure estimated worldwide [24] and with the percentage of adult smokers in Italy (about 20%) [32]. This result is also still of concern because a study we conducted ten years ago (academic year 2007–2008) in the same area and on schoolchildren of the same age (5–11 years) reported a percentage very similar to that observed today, which was 45.1% of children exposed to ETS [24]. Consequently, the current finding indicates a failure of prevention interventions specifically dedicated to smoking cessation and the protection of non-smokers. At the same time, this result clearly indicates the need for new, different, and more effective prevention strategies devoted to smoking cessation. On the contrary, a positive result emerged from the comparison of the application of the home smoking ban in the two different periods: ten years ago, in 38.4% of the monitored homes, any smoking ban was effective [24], that is more than twice compared to what we found in the present study (18.7% of homes without any smoking ban).

Another relevant result is related to the contribution of the educational level and the nationality of parents on the ETS exposure of children. In particular, we found that parents’ nationality does not significantly influence ETS exposure in domestic environments or the application of a partial or total smoking ban at home. On the contrary, Orton et al. reported an association between being of non-white ethnicity and the increase of ETS exposure at home [33]. However, the not-Italian parents participating in our research are of the same ethnicity of Italians. Typically, social determinants, such as nationality, socio-economic status, and income, are factors strictly related themselves and their lifestyles; thus, the role of these variables should be studied in depth to further understand their role on ETS exposure at home and the contribution of social predictors in applying a domestic smoking ban and home smoking policies to protect non-smoker cohabitants. In partial contrast with the results related to nationality, we found a significant influence of the educational level on the smoking dependence: lower maternal or paternal educational level were significant predictors of ETS exposure. This is in line with the results of previous studies performed in Italy [18] and in other countries [33,34,35,36], and it highlights the needs for specific smoking cessation interventions performed on smokers with a low educational level. These results were also confirmed by multivariate analyses, revealing that the lower education of the father or mother significantly increased the risk of children being exposed to ETS.

No differences were found in the application of a home smoking ban according to the educational levels; this result may be explained by the hypothesis that when an individual has a smoke addiction, she/he decides to smoke at home due to her/his level of nicotine dependence, independent of the level of education.

The modality of application of home smoking bans differs when the cohabitant smoker is only the mother (case 1), only the father (case 2), or when mother and father are both smokers (case 3). Indeed, we observed that a higher percentage of partial or total smoking bans were applied when the cohabitant smoker was the father (case 2), followed by case 1 (the cohabitant smoker was just the mother) and case 3 (both mother and father were smokers). The finding related to the case 2 can be explained by three reasons: (1) smoker fathers are more “sensitive” than smoker mothers to the potential domestic ETS exposure of non-smoker cohabitants; (2) the smoker fathers’ wives are less “permissive”, and they did not permit them to smoke at home; (3) on average, fathers stay at home for less time with respect to mothers and, consequently, they have less “opportunities” to smoke at home. The worst case was the home where both the mother and father smoked: more than 20% of children having both mother and father smokers lived in a domestic environment without any type of smoking ban, and the cohabitant smoker(s) also smoked at home in the presence of the children. Our results are in line with those reported by a systematic review on predictors of children’s secondhand smoke exposure at home: children whose mothers or both parents smoked were 2–13 times more likely to be exposed to ETS with respect to the other ones [36]. In contrast, Precioso et al. [37] carried out a survey in Portugal and found that children were more exposed to ETS in domestic environments when the smoker was the father.

## 5. Conclusions

The findings of the present study highlight that, despite the scientific evidence of a great number of health risks associated with ETS exposure, especially when those exposed are infants and children, the percentage of smokers that do not apply smoking bans in domestic environments is still high. Therefore, more and more effective preventive interventions are needed to protect children from exposure to passive smoking. Some of these interventions should be specifically dedicated to smokers with a low educational level and to smoker mothers that increase the risk of exposure of children at home.

## Figures and Tables

**Figure 1 ijerph-16-04062-f001:**
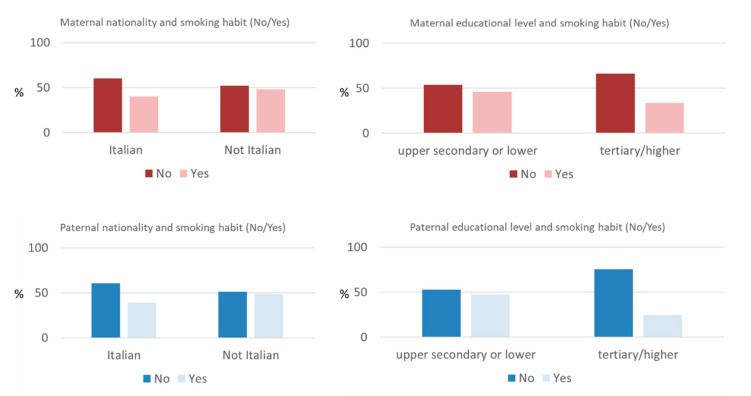
Association between paternal nationality and educational level and the prevalence of smoking habits. Statistically significant differences in the prevalence of smoking habits have been recovered both for different educational levels of mother or father (*p* = 0.022 and *p* < 0.001, respectively).

**Figure 2 ijerph-16-04062-f002:**
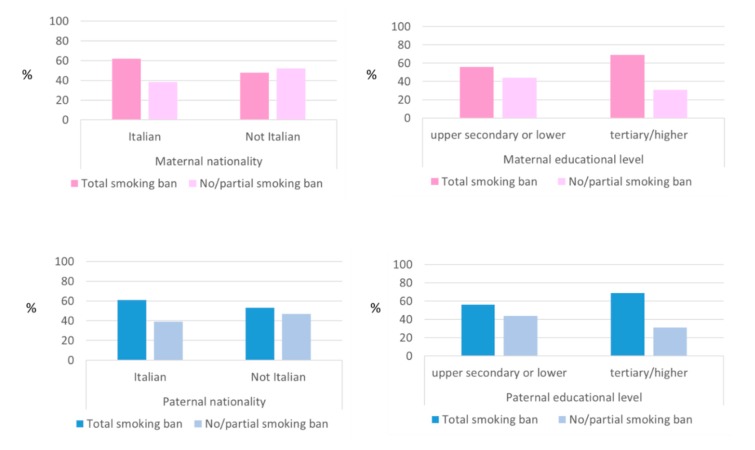
Association between maternal and paternal nationality or educational level and home smoking policies in children exposed to ETS.

**Figure 3 ijerph-16-04062-f003:**
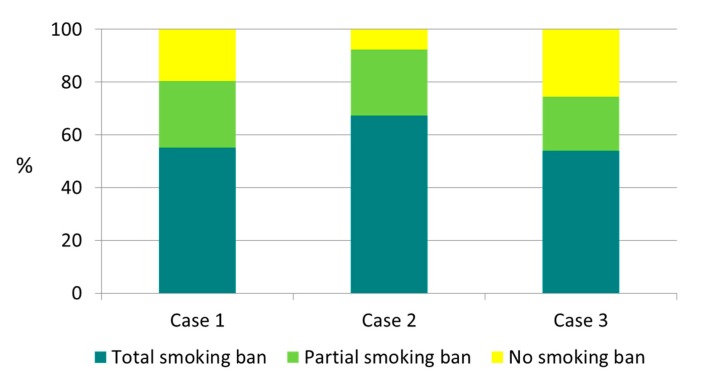
ETS exposure profile when the cohabitant smoker was only the mother (case 1), when the cohabitant smoker was only the father (case 2), and when mother and father were both smokers (case 3).

**Table 1 ijerph-16-04062-t001:** Characteristics of the study population and of the participants exposed to Environmental Tobacco Smoke (ETS).

Variable		Descriptives in % if not Stated Otherwise (*N*)	
		Whole Studied Population	Participants Exposed to Environmental Tobacco Smoke (ETS)
Age		Mean 8.24 (standard deviation 1.48)	Mean 8.19 (standard deviation 1.50)
Gender	Male	49.3 (*N* = 187)	47.7 (*N* = 74)
	Female	50.7 (*N* = 192)	52.3 (*N* = 81)
Maternal nationality	Italian	84.7 (*N* = 321)	81.3 (*N* = 126)
	Not Italian	15.3 (*N* = 58)	18.7 (*N* = 29)
Paternal nationality	Italian	86.8 (*N* = 329)	82.6 (*N* = 128)
	Not Italian	13.2 (*N* = 156)	17.4 (*N* = 27)
Maternal education (years)	Basic (≤9 years)	15.6 (*N* = 59)	20.0 (*N* = 31)
	Upper secondary (≤14 years)	47.8 (*N* = 181)	52.3 (*N* = 81)
	Tertiary/higher (≥17 years)	33.2 (*N* = 126)	27.1 (*N* = 42)
	Unknown	3.4 (*N* = 13)	0.6 (*N* = 1)
Paternal education (years)	Basic (≤9 years)	26.4 (*N* = 100)	34.8 (*N* = 54)
	Upper secondary (≤14 years)	45.1 (*N* = 171)	47.7 (*N* = 74)
	Tertiary/higher (≥17 years)	23.8 (*N* = 90)	14.2 (*N* = 22)
	Unknown	4.7 (*N* = 18)	3.3 (*N* = 5)
Exposure to ETS	Not exposed	58.9 (*N* = 224)	0 (*N* = 0)
	Exposed	41.1 (*N* = 155)	100 (*N* = 155)
Number of cohabitant smoker(s)	1	-	70.3 (*N* = 109)
	>1	-	29.7 (*N* = 46)
Home smoking policy	Total smoking ban	-	58.7 (*N* = 91)
	Partial smoking ban	-	20.0 (*N* = 31)
	No smoking ban	-	18.7 (*N* = 29)
	Unknown	-	2.6 (*N* = 4)
Number of cigarettes smoked in total/day		-	Mean 15.94 (standard deviation 9.71)
Number of cigarettes smoked at home/day		-	Mean 2.46 (standard deviation 4.20)
Number of cigarettes smoked at home in the presence of child/day		-	Mean 0.94 (standard deviation 2.85)

**Table 2 ijerph-16-04062-t002:** Logistic regression analysis of parental nationality and educational level associated with Environmental Tobacco Smoke (ETS) exposure among monitored schoolchildren.

Independent Variables		Adjusted ^1^ OR (95% CI)	*p*
Maternal nationality	Italian	0.844 (0.365, 1.953)	0.692
	Not Italian	Reference	
Paternal nationality	Italian	0.809 (0.310, 2.110)	0.664
	Not Italian	Reference	
Maternal education (years)	Upper secondary (≤14 years)	1.254 (1.098, 1.908)	0.046
	Tertiary/higher (≥17 years)	Reference	
Paternal education (years)	Upper secondary (≤14 years)	2.0603 (1.458, 4.647)	0.001
	Tertiary/higher (≥17 years)	Reference	

OR: Odds Ratio; 95% CI: 95% Confidence Interval; ^1^ Adjusted for all the four variables reported in the table as well as for gender and age; dependent variable: ETS exposure (not exposed to ETS was the reference category).

**Table 3 ijerph-16-04062-t003:** Logistic regression analysis of parental nationality and educational level associated with no/partial smoking ban at home among the subgroup of monitored schoolchildren exposed to Environmental Tobacco Smoke (ETS).

Independent Variables		Adjusted ^1^ OR (95% CI)	*p*
Maternal nationality	Italian	0.564 (0.155, 2.050)	0.385
	Not Italian	Reference	
Paternal nationality	Italian	0.783 (0.175, 3.495)	0.748
	Not Italian	Reference	
Maternal education (years)	Upper secondary (≤14 years)	1.656 (0.754, 3.636)	0.209
	Tertiary/higher (≥17 years)	Reference	
Paternal education (years)	Upper secondary (≤14 years)	0.861 (0.332, 2.231)	0.758
	Tertiary/higher (≥17 years)	Reference	

OR: Odds Ratio; 95% CI: 95% Confidence Interval; ^1^ Adjusted for all the four variables reported in the table as well as for gender and age; dependent variable: home smoking ban (total smoking ban at home was the reference category).

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
