# Peer review of "Further Insights on Predictors of Environmental Tobacco Smoke Exposure during the Pediatric Age"

_ijerph, 2019, doi:10.3390/ijerph16214062_

Round 1

Reviewer 1 Report

This manuscript describes potential exposure to passive smoke in the homes of children in one area of Italy.  It is a cross sectional survey, which focuses on exposure without health outcomes.  The children were recruited via schools, and all ethical contraints were met.

However, the data analysis is rather simplistic and does not really give information on the specific subgroups on which to focus smoking cessation programs (or at least programs to discourage smoking in the home).  The investigators could consider multivariate prediction models to identify important subgroups based on nationality, sex, age and educational attainment of the parents.  The outcome could be categorical (i.e. multinomial logistic regression) to predict home smoking policy.   Such a model could also predict smoking vs non-smoking.

Further, tables 1 and 2 can be combined to give a better comparison of non-smoking vs smoking parents.   The figures can be improved using the models as then the proportions of smoking would be controlled for other important variables.

Such improvements to the analysis would give better data as to where to target relevant programs.

Author Response

We thank the reviewer for her/his constructive comments and suggestions on our manuscript.

R1: This manuscript describes potential exposure to passive smoke in the homes of children in one area of Italy. It is a cross sectional survey, which focuses on exposure without health outcomes. The children were recruited via schools, and all ethical contraints were met. However, the data analysis is rather simplistic and does not really give information on the specific subgroups on which to focus smoking cessation programs (or at least programs to discourage smoking in the home). The investigators could consider multivariate prediction models to identify important subgroups based on nationality, sex, age and educational attainment of the parents. The outcome could be categorical (i.e. multinomial logistic regression) to predict home smoking policy. Such a model could also predict smoking vs non-smoking.

A: According to the reviewer’s comment, we performed two further statistical analyses: we run two regression models, the first one to assess the association between parental nationality and educational level with ETS exposure among monitored schoolchildren and the second one to evaluate parental nationality and educational level associated with no/partial smoking ban at home among the subgroup exposed to ETS. The results were reported in Table 2 and 3 of the R1 version and discussed in the revised text.

R1: Further, tables 1 and 2 can be combined to give a better comparison of non-smoking vs smoking parents.

A: As suggested by the reviewer, we combined Table 1 and 2.

R1: The figures can be improved using the models as then the proportions of smoking would be controlled for other important variables.

A: We reported the results of the multivariate analyses in Table 2 and 3.

R1: Such improvements to the analysis would give better data as to where to target relevant programs.

A: Thank you again for suggestion and comments.

Reviewer 2 Report

In this manuscript the authors report on a cross-sectional study of smoking habits of parents. The authors correctly state that knowledge of factors that contribute to children’s exposure to ETS is important to manage risk and develop effective interventions. Unfortunately, factors assessed in this study were limited to mother or father, education level and nationality so the study is not really adding any new insight into this important issue. However, the information are still of some interest as it highlights that, in some places, there is still a long way to go in reducing children’s exposure.

Specific comments:

1. Aim 1 needs to be rewritten as the study does not investigate the contribution of the factors on ETS exposure. It just identifies factors that seem to be important. The actual ‘contribution’ of these factors is not assessed.

2. The Introduction is too long. The explanation of SHS and THS can be shortened (as this was not assessed), as can discussion on children’s vulnerability (eg remove lines 61/62)

3. No multivariate analyses were conducted. Using logistic regression will at least provide some measure of ‘contribution’ (ie odds of being exposed with lower education)

4. I think the figures are unnecessary and the data are probably presented as text. Fig 1 is a bit misleading as there does seem to be a difference between Italian and non-Italian parents. Obviously this difference is not significant but it would be easier to see numbers.

5. Discussion

5.1: 40% in this sample is higher than the national average (as reported in Lugo et al. Tumori 2017; 103: 353 – 59). What might be the reason for this?

5.2: The trend in national smoking rates should be considered in the discussion on lack of change over a 10-year period (lines 210/11).

5.3: lines 215 – 218 need to be rewritten as it doesn’t make sense at the moment. Do the authors mean that there is a two-fold increase in the number of homes with a smoking ban in the current study compared to 10 years previously?

5.4: when discussing the lack of (statistical?) difference between Italian and non-Italian parents it may be worth adding that smoking rates in the countries of origin may be similar to Italian national rates - if that is the case).

5.5: Lines 233 – 236, is there any evidence to support the ‘fact’ that smoke addiction affects the decision to smoke inside home independent of education level. I would suggest that education level would still be important irrespective of the level of addiction, so it would be good to have evidence to support the statement (or change the word ‘fact’)

5.6: lines 242, fathers are more ‘sensitive’ than who?

6. The manuscript needs some editing. Some of the main grammatical errors are highlighted below (but the list is not exhaustive)

line 33, change ‘associated to …’ with ‘associated with…’ change ‘researches’ to ‘research’ change ‘evidences’ to evidence’ use words such as ‘found’ or ‘observed’ rather than ‘recovered’ when discussing study findings change ‘/die’ to ‘/day’ change ‘statistical elaboration’ to ‘statistical analyses’ change line 114 to ‘…they were asked to indicate the number of cohabiting smokers’ change line 117 to ‘ .. number of cigarettes smoked in total, at home generally, and at home when child is present

Author Response

R2: In this manuscript the authors report on a cross-sectional study of smoking habits of parents. The authors correctly state that knowledge of factors that contribute to children’s exposure to ETS is important to manage risk and develop effective interventions. Unfortunately, factors assessed in this study were limited to mother or father, education level and nationality so the study is not really adding any new insight into this important issue. However, the information are still of some interest as it highlights that, in some places, there is still a long way to go in reducing children’s exposure.

A: We thank the reviewer for her/his constructive comments and suggestions on our manuscript.

R2: Specific comments:

R2: 1. Aim 1 needs to be rewritten as the study does not investigate the contribution of the factors on ETS exposure. It just identifies factors that seem to be important. The actual ‘contribution’ of these factors is not assessed.

A: We changed the aim as follows: “to investigate the association between several factors and ETS exposure among a group of healthy children living in Italy and….

R2: 2. The Introduction is too long. The explanation of SHS and THS can be shortened (as this was not assessed), as can discussion on children’s vulnerability (eg remove lines 61/62).

A: Based on the reviewer’s comment, explanation of SHS and THS and discussion on children’s vulnerability were shortened.

R2: 3. No multivariate analyses were conducted. Using logistic regression will at least provide some measure of ‘contribution’ (ie odds of being exposed with lower education).

A: According to the reviewer’s comment, we performed two further statistical analyses: we run two regression models, the first one to assess the association between parental nationality and educational level with ETS exposure among monitored schoolchildren and the second one to evaluate parental nationality and educational level associated with no/partial smoking ban at home among the subgroup exposed to ETS. The results were reported in Table 2 and 3 of the R1 version and discussed in the revised text.

R2: 4. I think the figures are unnecessary and the data are probably presented as text. Fig 1 is a bit misleading as there does seem to be a difference between Italian and non-Italian parents. Obviously this difference is not significant but it would be easier to see numbers.

A: In our opinion, also based on the scope of the IJERPH and the published papers, it is more useful to maintain Figures rather than describe an excess of numbers and details in the text. For this reason, we ask the reviewer to agree with our choice.

R2: 5. Discussion

R2: 5.1: 40% in this sample is higher than the national average (as reported in Lugo et al. Tumori 2017; 103: 353 – 59). What might be the reason for this?

A: Our percentage was calculated based on at least one cohabitant smoker; thus, if on average one child has two parents, 40 children exposed to parental smoking every 100 children (= 200 parents) correspond to 40 smoker parents that is 20% of all parents. This results is in line with the paper of Lugo et al. that we cited in the revised text.

R2: 5.2: The trend in national smoking rates should be considered in the discussion on lack of change over a 10-year period (lines 210/11).

A: According to the paper of Lugo et al., smoker prevalence among Italian adult population decreased about 1.5% (from 22.8 to 21.4%). In our manuscript, ETS exposed children are about 40% while in our previous study they were 45%. This difference, in terms of effectiveness of prevention interventions, corresponds to a failure from a public health point of view, as we highlighted in the text.

R2: 5.3: lines 215 – 218 need to be rewritten as it doesn’t make sense at the moment. Do the authors mean that there is a two-fold increase in the number of homes with a smoking ban in the current study compared to 10 years previously?

A: Based on the reviewer’s suggestion, we modified the sentences as follows: “On the contrary, a positive result emerged from the comparison of the application of the home smoking ban in the two different periods: ten years ago, in 38.4% of the monitored homes any smoking ban was effective [24], that is more than twice compared to what we found in the present study (18.7% of home without any smoking ban).”.

R2: 5.4: when discussing the lack of (statistical?) difference between Italian and non-Italian parents it may be worth adding that smoking rates in the countries of origin may be similar to Italian national rates - if that is the case).

A: It is very difficult to indicate the smoking rates in all the countries of origins because the not Italian parents had the same ethnicity of Italian ones, but they came from many different European countries (Albania, Poland, Romania, Moldova and other eastern Europe countries).

R2: 5.5: Lines 233 – 236, is there any evidence to support the ‘fact’ that smoke addiction affects the decision to smoke inside home independent of education level. I would suggest that education level would still be important irrespective of the level of addiction, so it would be good to have evidence to support the statement (or change the word ‘fact’)

A: The fact that the nicotine addiction level could influence the adoption of a smoking ban was an authors’ hypothesis. Thus, we changed the text accordingly.

R2: 5.6: lines 242, fathers are more ‘sensitive’ than who?

A: Than the smoker mothers. We added this clarification in the text.

R2: 6. The manuscript needs some editing. Some of the main grammatical errors are highlighted below (but the list is not exhaustive)

line 33, change ‘associated to …’ with ‘associated with…’ change ‘researches’ to ‘research’ change ‘evidences’ to evidence’ use words such as ‘found’ or ‘observed’ rather than ‘recovered’ when discussing study findings change ‘/die’ to ‘/day’ change ‘statistical elaboration’ to ‘statistical analyses’ change line 114 to ‘…they were asked to indicate the number of cohabiting smokers’ change line 117 to ‘ .. number of cigarettes smoked in total, at home generally, and at home when child is present

A: We thank the reviewer for her/his suggestions on grammatical errors. The authors, with the help of a native English speaker, revised the manuscript for main grammatical errors.

Round 2

Reviewer 1 Report

The authors have responded adequately to the previous comments.  However, the manuscript still needs substantial editing for English language and syntax.

Author Response

We thank the reviewer for her/his constructive previous comments and suggestions on our manuscript. The manuscript has been revised for English language and syntax by a native English speaker.

Reviewer 2 Report

The authors have addressed my concerns. One minor point, the following sentence should be changed from '...this result can be explained by the hypothesis that when an individual have a smoke addiction.......' (lines 254 - 257) to '... this result may be explained by the hypothesis that when an individual have a smoke addiction...'

Author Response

We thank the reviewer for her/his constructive previous comments and suggestions on our manuscript. We changed the sentence as suggested by the reviewer. In addition, the manuscript has been revised for English language and syntax by a native English speaker.